# Potential Challenges of the Extraction of Carotenoids and Fatty Acids from Pequi (*Caryocar brasiliense*) Oil

**DOI:** 10.3390/foods12091907

**Published:** 2023-05-06

**Authors:** Camila Rodrigues Carneiro, Adamu Muhammad Alhaji, César Augusto Sodré da Silva, Rita de Cássia Superbi de Sousa, Simone Monteiro, Jane Sélia dos Reis Coimbra

**Affiliations:** 1Department of Chemistry, Universidade Federal de Viçosa, Viçosa 36570-900, Brazil; 2Department of Food Technology, Universidade Federal de Viçosa, Viçosa 36570-900, Brazil; 3Department of Food Science and Technology, Faculty of Agriculture and Agricultural Technology, Kano University of Science and Technology, Wudil 713101, Nigeria; 4Department of Mechanical Engineering, Faculty of Technology, University of Brasilia, Brasilia 70910-900, Brazil; 5Graduate Program of Chemical Engineering, Federal University of Goias, Goiania 74690-900, Brazil

**Keywords:** artisanal extraction, bioactive compounds, carotenoid composition, fatty acid profile, solvent extraction, pigments

## Abstract

Pequi is a natural source of bioactive compounds with wide versatility for fresh or processed fruit consumption, but it is still little explored economically. Functional foods are the subject of diverse scientific research since, in addition to being nourishing, they contain bioactive compounds capable of promoting several benefits to the human body. Pequi is a fruit species native to the Brazilian *Cerrado*, which is rich in oil and has components with a high nutritional value, such as unsaturated fatty acids (omega-3, omega-6, EPA, and DHA), antioxidants (carotenoids and phenolic compounds), and vitamins. Therefore, the present narrative review aims to compile and critically evaluate the methods used to extract oil from the pulp and almonds of pequi and describes the carotenoid separation from the oil because carotenoids are natural pigments of great interest in the pharmaceutical and food industries. It is emphasized that the main challenges linked to bioactive compound extraction are their susceptibility to degradation in the processing and storage stages of pequi and its derived products.

## 1. Introduction

The Brazilian *Cerrado* is a tropical climate biome with low vegetation marked by shrubs, trees, and grasses coexisting with a gently undulating relief and great water resources on acidic soil. The climatic and soil conditions of the *Cerrado* favor the development of several fruit species with great potential for agricultural and technological use. The pequi (*Caryocar brasiliense*) tree grows with a leafy crown of up to 12 m. Its fruit is used (i) in folk medicine to combat aging, prevent and treat vision-related diseases, and as an anti-inflammatory, healing, and gastroprotective agent, among other applications [1]; (ii) as an ingredient in several recipes such as rice and chicken; (iii) as a raw material for food products such as ice cream, liqueurs, flour, and animal feed [2]; (iv) in the food industry as an ingredient and substitute for artificial dyes; and (v) in the pharmaceutical and cosmetic industries to formulate medicines and skin creams. Pequi harvesting and processing are income sources for many families; thus, the fruit is economically relevant for agro-extractive populations and traders in some Brazilian provinces [1,2]. The oil extracted from pequi pulp and almonds (i) is rich in bioactive compounds such as carotenoids, fatty acids, and vitamins A, C, and E and (ii) presents an orange color, a mild aroma, and the characteristic flavor of the fruit, being the most identifiable feature of pequi [1]. Thus, preserving the integrity of the bioactive compounds of pequi fruits can ensure the maintenance of their technological and functional characteristics [3].

An unmistakable trend in the last decade is the adoption by consumers of healthier lifestyles, focusing on healthy eating and leading an active routine [4]. With the rapid development of the food industry, consumers have exhibited new expectations regarding food and healthful diets [5] since food intake highly influences the development of some autoimmune disorders and illnesses, such as cancer, nephropathies, and diabetes. Thus, including foods of high nutritional value containing bioactive compounds may be a way to prevent diseases [5,6]. For instance, due to the recent spread of the COVID-19 pandemic in 2020 and the consequent amendments in lifestyle (staying at home or exercising outside and at home), consumers have changed their eating and dietary supplementation habits [7,8]. Furthermore, the link between individuals with obesity and the significant increases in morbidity and mortality from COVID-19 has raised consumers’ concerns regarding bodyweight [8,9]. Therefore, food sector companies must keep up with consumers’ interests and requirements when designing novel products.

Since carotenoids play an imperative role in boosting body immunity, mainly via their effects on several antioxidant and anti-inflammatory pathways and their components affecting immune response, they can act as immune enhancers against COVID-19 and other emerging diseases and related syndromes [10]. Yi et al. [10] reported that both the dietary intake and circulating levels of β-carotene are inversely associated with the risk of all-cause mortality. This is mainly the case for dietary β-carotene, which has a protective effect on preventing non-communicable chronic diseases. The epidemiologic study of Ferreira et al. [11] reported the association between carotene and health outcomes, including Alzheimer’s disease, fracture, and various types of cancers.

In addition to its high carotenoid content, pequi oil provides unsaturated fatty acids, which benefit human health and nutrition. Among them, polyunsaturated fatty acids (PUFAs), such as linolenic and linoleic acids, are essential because humans cannot produce them; as such, they must be obtained from food [12]. Additionally, oils with a high content of monounsaturated fatty acids, such as oleic acid, can lower harmful cholesterol levels and protect against heart disease [13].

Considering the various technological processes available, processing pequi fruit and its oil is a great alternative to produce healthier foods combined with the sustainable development of the *Cerrado* biome. In this context, the present narrative review highlighted the following: the potential of the utilization of the pequi fruit, a species rich in bioactive compounds; the pulp and almond oil separation from pequi using the artisanal, mechanical, solid–liquid, and supercritical methods that affect carotenoid and fatty acid extraction; the processing line of pequi oil separation; the influences of the oil extraction methods on the pigment and the fatty acid contents; the separation of carotenoids and fatty acids from pequi oil; the challenges of obtaining bioactive compounds from pequi fruit; and future perspectives on the valorization of pequi and its derivatives.

## 2. The Brazilian *Cerrado* Biome

Brazilian biodiversity accounts for 30% of the animal and plant species on the planet and is distributed in six major biomes: the *Cerrado*, the southern fields and forests, the Atlantic Forest, the Caatinga, the Amazon Forest, and the Pantanal. The typical vegetation of the *Cerrado* is similar to that of the savannah, predominantly made up of low trees (up to 20 m in height), shrubs, and grasses. These trees have crooked trunks, twisted branches, thick bark, and thick leaves, mainly due to the deficit of micronutrients in the soil, such as aluminum [14]. One of the great highlights of the *Cerrado* is its native fruit species, such as pequi (*Caryocoar brasiliense*), macaúba (*Acrocomia aculeata*), and tucumã (*Astrocaryum aculeatum*), among many others. These fruits present high nutritional value and attractive sensory characteristics, such as color, flavor, and aroma, although they are still little explored economically [14,15]. Pequi stands out for its nutritional content and therapeutic potential.

## 3. Pequi

Pequi belongs to the Caryocaraceae family of the genus *Caryocar*. The name pequi originates from the indigenous *Tupi* culture and means thorny bark, referring to the thorns of the fruit’s endocarp [16]. It is also known as piqui, piquiá, piqui-do-cerrado, piquiá bravo, pequerim, thorn almond, horse grain, and suarí.

Pequi is a quadrupoid fruit (Figure 1) with a greenish-brown exocarp and two mesocarps. A white pulp forms the external mesocarp, and a light yellow to dark orange pulp forms the internal mesocarp. The latter is the edible fruit portion inside that covers a brown and spiny endocarp [16]. It consists of approximately 84% husk, 6% endocarp, 8% pulp, and the remaining 2% is almond, totaling an average mass of 120 g. The average height of the pequi is 5.8 cm, and the diameter varies between 5.54 cm and 6.48 cm [17].

The chemical composition of pequi can also change according to the part of the fruit (pulp or almonds, for instance). Thus, Table 1 presents the pequi composition of pulp and almond to compile variations regarding the macronutrient and micronutrient constituents. The pulp and almonds are rich in proteins, lipids, and dietary fibers, such as pectin (rhamnogalacturonan) and hemicellulose (arabinogalactans, xylans, and glucomannans). In addition, they have a high percentage of carotenoids, phenolic compounds, vitamin C (ascorbic acid), and tocopherols, which are bioactive components with an expressive antioxidant capacity for both free radical scavenging and oxidation reduction [18]. The phenolic content of the pulp (209 mg/100 g) is higher than that of other fruit pulps, such as açaí (*Euterpe oleracea*), with a phenolic content of 136.8 mg/100 g; guava (*Psidium guava*), with a phenolic content of 83.1 mg/100 g; and strawberry (*Fragaria vesca*), with a phenolic content of 132.1 mg/100 g. Therefore, pequi fruits and their residues are an excellent natural source of extractable bioactive compounds of industrial interest for producing medicines and functional foods [2,16].

Most lipids from pequi comprise unsaturated fatty acids, which are great allies in preventing cardiovascular diseases. The profiles of fatty acids, and their average percentages, of the pulp and almonds from pequi cultivated in different Brazilian regions are listed in Table 2. Oleic and palmitic fatty acids stand out from other fatty acids in smaller proportions. The cultivation region also interferes with the fatty acid percentage since there are different types of fatty acids according to the country’s regions [17].

Carotenoids, another highly valued constituent of pequi pulp, are fat-soluble pigments responsible for the yellowish fruit color. They are classified into carotenes or xanthophylls according to their chemical structure. Carotenes have a linear or cyclic hydrocarbon chain at one or both ends of the molecule. Xanthophylls comprise the oxygenated derivatives of carotenes, for which groups include hydroxyl (β-cryptoxanthin), keto (canthaxanthin), epoxide (violaxanthin), and aldehyde (β-citraurine). Carotenoids exhibit antioxidant properties beneficial for human health by preventing cancer, cardiovascular illnesses, and vision diseases [18] and others described in reference [22].

Clinical and epidemiological studies have demonstrated that lutein and zeaxanthin xanthophylls prevent macular degeneration and cataracts [22]. Lycopene and β-carotene pigments can prevent skin and prostate cancers and cardiovascular diseases by delaying LDL cholesterol oxidation and forming atherosclerosis plaques [22]. The consumption of foods rich in carotenoids has been associated with other health benefits, such as the maintenance of cognitive functions, a reduction in depressive symptoms, and the prevention of osteoporosis in elderly individuals, via the protection of bone mineral density [22,23]. Thus, pequi, similarly to other fruits of the *Cerrado* biome, can be used as a raw material for the formulation of herbal medicines and functional foods [24,25].

Phenolic compounds of pequi are also valuable constituents. They are secondary metabolites that contribute to sensory aspects (astringency, color, and fruit aroma) and the protection against microbiological, thermal, and oxidative damage [24]. Regarding the functional attributes applicable to preventing and treating several diseases, phenolic extracts from pequi have antioxidant, antidiabetic, anti-inflammatory, antimicrobial, anticarcinogenic, analgesic, and cost-protective actions [18,19,24]. The crude extract of pequi has significant contents of gallic acid, ellagic acid, and p–coumaric acid [25]. Even its fruit residues are rich in phenolic compounds, such as pequi peel flour, with high concentrations of gallic acid [2,26].

Therefore, obtaining phenolics from pequi fruit and its derivatives is an excellent way to show the value of the fruit and its production chain. These biocompounds can also act in the structural and functional quality of corn starch-based biofilms and in the formulation of cosmetics to fight age and treat skin pigmentation disorders [27,28,29].

Different methods can be applied to separate phenolic compounds from fruits, such as solvent extraction, supercritical extraction, microwave-assisted extraction, membranes, ultrasound, or high hydrostatic pressure [25,28]. However, extraction with organic solvents is more frequently used than other techniques due to its efficiency and lower operating cost [28]. For solvent extraction, renewable or low-toxicity solvents may be prioritized to ensure environmental and operational safety [28,30].

The most commonly used technique for the identification and quantification of carotenoids is ultraviolet-visible spectrophotometry (UV-Vis) [31], and for phenolic compounds, it is high-performance liquid chromatography (HPLC) [32]. The main carotenoids and phenolic compounds identified in pequi fruit and some of its derivatives are listed in Table 3. Variations in the contents of carotenoids and phenolics may be due to the differences in the cultivation region and the maturation stage of the fruits [15,33].

## 4. Pequi Oil

Pequi oil may be obtained from the mesocarp (pulp) or the almonds. According to De Lima et al. [17], the pequi oil contents of the pulp and almonds are approximately 33.4% and 51.5%, respectively. The pequi oil has an orange color, a mild aroma, and the characteristic flavor of the fruit. Its high nutritional quality is due to the high percentage of carotenoids, vitamin C, phenolic compounds, and unsaturated fatty acids, mainly oleic and palmitic acids [36]. Furthermore, pequi oil is used in folk medicine to prevent heart, eye, and respiratory diseases [42]. The medicinal applicability of pequi oil has been reported in the literature.

Miranda-Vilela et al. [3] developed a process to obtain gelatinous capsules from the extracts of pequi pulp with antioxidant potential and antimutagenic properties. The study confirmed the therapeutic efficacy of the product with 126 athletes before and after a marathon. These athletes showed less muscle inflammation, less damage to the DNA of cells, and lower oxidative stress after ingesting capsules, explaining the efficacy of the pequi derivatives. Moreover, the authors described that pequi oil has the potential to help reduce blood pressure in athletes. Colombo et al. [42] evaluated the effects of oral supplementation with pequi oil in cancer treatment and reported that the ingestion of the oil reduced DNA damage and lipid peroxidation in a model of lung carcinogenesis chemically induced in vivo. Miranda-Vilela et al. [36] verified the effectiveness of pequi oil in reducing tumor growth, potentiating the anticancer effects of magnetic hyperthermia therapy, and alleviating the side effects from chemotherapy in animal breast cancer models. Torres et al. [37] described that pequi almond oil could delay the oxidative process triggered by free radicals and improve the hepatotoxic and anti-inflammatory protections of the body since it contains polyunsaturated fatty acids in high content, essential fatty acids (ω-3 and ω-6) and other bioactive compounds. Batista et al. [43] reported the acceleration of the healing process of lesions in animals by using a cream based on pequi oil. Due to its chemical properties, aroma, and low melting temperature (37 °C), pequi oil is widely used as a culinary ingredient, an input for producing lubricants and biofuels, and in the cosmetic and pharmaceutical industries [3].

## 5. Pequi Oil Extraction Methods

Considering its broad employability, it is desirable to establish the operating conditions of the different processing methodologies that allow pequi oil to be extracted efficiently, with time and cost savings [44]. Table 4 lists the techniques commonly used to separate oil from pequi fruits, the operating conditions, and the extractive yields. The techniques are artisanal extraction, mechanical extraction, solid–liquid extraction, and supercritical extraction. The solid–liquid and supercritical extraction methods revealed the highest extractive yields because of the specificity of the methodologies and the use of more sophisticated equipment. However, to date, such methodologies are mainly applied on a laboratory scale in studies for the characterization and medicinal application of pequi oil [18,37].

### 5.1. Artisanal Extraction

The artisanal extraction of pequi oil is an ancient technique. This process is carried out with fruits in natura subjected to cooking and aqueous extraction using boilers, such as pans, paint cans, or even an old refrigerator compartment [47]. It involves a series of unit operations of separation and heat exchange and is adopted by many families living in economically vulnerable conditions in the *Cerrado*. In this sense, due to its economic and social significance, the technique has great potential to be recognized as part of Brazil’s intangible cultural heritage. Because sophisticated equipment or methodologies are not used, the productivity, yield, and oil quality are low compared to other techniques. According to Aquino et al. [48], for the artisanal extraction of 1 L of pequi, approximately 50 dozen fruits and 10 h of work are needed. Despite the lower oil extraction yield compared to other methods, the pequi oil obtained by hand has a composition rich in nutrients and greater stability during storage at room temperature [45].

A flowchart of the artisanal process of extracting pequi oil is shown in Figure 2. The artisanal extraction is illustrated in Figure 3.

Figure 3a represents the beginning of the artisanal process to obtain pequi oil by peeling the fruits and removing the exocarps. Harvesting the pequi from the ground is recommended to ensure fully ripe fruits. Next (Figure 3b), the pequi is placed in pans or large pots, added to water, and taken to the cooking process for approximately 40 min to facilitate its disintegration, cooling, and removal from the cooking water. Then (Figure 3c), cooked and cooled fruits are brought to manual maceration using pestles in a large wooden container called a trough. Work is carried out during periods of the day when the ambient temperature is low. During maceration, the pulp and almonds are separated, and the transfer of flavor, odor, and other characteristics of the pequi to the oily extract is maximized. The cooled water is added to facilitate the pestles’ action and indicates the end of the process. The water favors the coagulation of fat that migrates to the upper part of the container and separates from the oil via decanting. Figure 3d shows the evaporation of the residual water from the mixture. The oil remains in the pan and is then filtered and bottled.

### 5.2. Mechanical Extraction

The mechanical extraction of pequi oil follows a conventional route for extracting vegetable oils characterized by applying compressive forces to separate the liquids from the solids [49]. Castanheira [50] patented a process to obtain pequi oil on an industrial scale by the mechanical pressing of clean and peeled fruit, followed by the physical separation of the oil, pulp, and thorn components, as represented in Figure 2. The initial part of this mechanical processing line is similar to the artisanal method. The press can be of the hydraulic or continuous type. The collected fruits are sanitized and peeled to remove surface dirt and skin. Then, fruit cooking is performed under at 80 °C for 3 min to soften the fleshy part. Decoction favors pressing and increases the oil extraction yield. Next, part of the cooking water is removed by decanting. The pequi is pressed, and the cake is separated [50].

The cake contains the oil, pulp, and thorns of the fruit. Centrifugation is used to physically separate these components. In this process, the spines produce the sedimented phase, and the oil and pulp make up the supernatant phase. The supernatant is removed and filtered to separate the oil from the pulp. Then, the oil is concentrated by removing the water by evaporation, increasing its shelf life [50,51]. Compared to artisanal extraction, oil extraction by pressing has advantages such as equipment and techniques that allow a higher extractive yield in less processing time and a greater preservation of its natural properties [49]. Combining two extractive techniques, such as pressing and solvent extraction, increases the process efficiency and allows the extraction of residual oil from the pulp cake in the filtration step [49,50].

### 5.3. Solid–Liquid Extraction

The operating principle of solid–liquid extraction consists of oil extraction from pequi pulp or kernels in contact with a nonpolar liquid solvent. The formed solid phase is separated from the oily liquid phase. The oil is separated from the solvent, and the solvent can be recovered and recycled [30].

Before starting the extraction process, the steps of drying, grinding, or cooking can be carried out to facilitate solvent penetration and the solubilization of the oil molecules. After this preliminary step, the solvent and solid fruit matrix are maintained in contact to promote oil migration from the solid to the solvent phase due to the greater affinity between the oil and the solvent. An equilibrium is reached after the formation of two phases, which are separated. The oil + solvent-rich phase is called the extract, and another residual solid phase, which is poor in oil, is called the raffinate. Next, the phases are then separated by, for example, filtration or settling [30]. After phase separation, the extracting solvent must be removed from the oil because most solvents have a high cost and a certain degree of toxicity that is harmful to health and the environment. This removal process is usually performed by distillation, and the recovered solvent can be reused. The described steps are represented in Figure 2.

Pequi oil extraction, similarly to oil extraction from other types of vegetables, is carried out mainly using hexane as a solvent due to the low polarity, selectivity, and narrow boiling range of hexane, making its interaction with the oil easier and the extraction faster [51]. However, hexane presents operational risks to human health and the environment, mainly due to its toxicity and flammability. These solvent characteristics have driven the replacement of hexane with renewable solvents, such as less toxic ethanol [30], and other organic solvents that offer lower operational risks, such as acetone [18]. 

The efficiency of the oil extraction process depends, in addition to the type of solvent used, on operational parameters such as temperature, agitation frequency, and contact time, which can affect the mass transfer between the solid and liquid phases [52]. For instance, high temperatures favor the kinetics and yield of extraction. However, as carotenoids and other bioactive compounds are sensitive to light and heat, extraction should be performed at milder temperatures to preserve the natural components of the oil [53]. Table 5 lists some pequi oil solid–liquid extraction conditions reported in the literature. Compared to hexane and other organic solvents, ethanol performs oil extraction satisfactorily. Thus, using a solvent from a renewable source can be an interesting way to increase the safety and sustainability of the extractive process of vegetable oils.

### 5.4. Supercritical Fluid Extraction

Supercritical extraction uses fluids in supercritical conditions to obtain pequi oil. Supercritical fluids (SCF) present properties such as density, diffusivity, and viscosity and characteristics of gases and liquids simultaneously, conferring greater capacity to fluids to act as solvents. SCF selectivity can be increased under variations in operational parameters, such as temperature and pressure, determining the optimal conditions for extracting specific vegetable oils. The SCF most commonly utilized in vegetable oil extraction is CO_2_ due to its ability to easily extract lipophilic compounds, its low cost, its nontoxic and nonflammable properties, and its easy removal of the extracted products [18,46]. Table 6 presents the CO_2_ and subcritical propane conditions used for oil extraction from pequi pulp.

Regarding the extraction of pequi oil, the supercritical method is promising as it has good efficiency; does not require high extraction temperatures, which help to preserve the oil’s bioactive compounds, such as carotenoids and fatty acids; produces an oily extract with high purity; and promotes easy separation between the solute and SCF [18]. 

Table 7 shows the advantages and disadvantages of the different methods of pequi oil extraction.

## 6. Bioactive Compounds of Pequi Oil

Vegetable oils play a vital role as food components as they are sources of natural antioxidants and other essential components for the human body. In addition to its nutritional value, pequi oil contains bioactive compounds, such as carotenoids, fatty acids, vitamins, and phenolic compounds, that arouse interest due to their therapeutic properties as anticancer, antiviral, anti-inflammatory, and hypoglycemic agents, for example. Obtaining and concentrating bioactive compounds from pequi oil aims to increase the availability of these compounds in the market. The bioactive components of pequi oil in most quantities are unsaturated fatty acids (57.87 ± 0.006 g/100 g) and carotenoids (33 mg/g). Pessoa et al. [18] also showed the high amounts of phytosterols in pequi pulp oil, including γ-tocopherol (26.24 µg/g of oil), β-sitosterol (10.22 mg/g of oil), and α-tocopherol (9.49 µg/g of oil).

### 6.1. Carotenoids

Carotenoids are naturally red, yellow, or orange pigments found in plants, algae, bacteria, fungi, and some materials of animal origin. Their expression in plant oils occurs due to the interaction between hydrophobic molecules and cell lipids [15]. The valorization of carotenoids as a natural dye in the food and pharmaceutical industries is due to the feasibility of their use to replace harmful synthetic dyes for human health. In the cosmetic segment, emulsions of pequi oil in the products enhance the antioxidant and skin protection properties and prevent lipid peroxidation that can delay skin aging because of the formation of free radicals. 

Ribeiro et al. [45] described that β-carotene is the primary carotenoid in pequi oil, corresponding to 50% of its composition. Moreno et al. [58] found more carotenoids in pequi pulp oil, nearly four times (549.84 μg of β-carotene/g of pequi oil) that found in pequi pulp (137.6 μg of β-carotene/g of pulp). Extractions performed at low temperatures favored obtaining an enriched β-carotene oil fraction due to the thermosensitivity of the pigment. Different methods for carotenoid extraction are described in the literature, such as saponification, solvent extraction, enzyme-assisted extraction, ultrasound- and microwave-assisted extraction, and supercritical fluid extraction [59,60].

The method most applied to extract carotenoids from pequi oil is solvent extraction due to the efficiency, low energy cost, simplicity of implementation in a laboratory on a small scale, and the possibility of execution at low temperatures, avoiding the thermal degradation of carotenoids in the oil [61]. Thus, choosing a suitable solvent is also essential as this component affects the efficiency of the process and can influence the stability of the pigments and the toxicity of the final product [62]. Some solvents used for carotenoid extraction from pequi oil are acetone (10.93 mg/g and 23.86 mg/g of extract), water (0.25 mg/g of extract), and ethanol (0.56 mg/g of extract) [53,62,63]. Acetone presents the highest extraction efficiency among the reported solvents; however, it is polluting and toxic, compromising safety and adding environmental problems to the process. To overcome this limitation, extraction with more environmentally friendly solvents is suggested. Among them, ethanol stands out due to its low cost, affinity with hydrophobic compounds, and ability to extract polar and nonpolar carotenoids efficiently [60,61]. Thus, ethanol and other renewable compounds with similar characteristics have the potential to be used, alone or in combination with other solvents, to make carotenoid extraction more economical and sustainable [30].

Similarly, oils extracted with hexane and ethanol showed more significant peroxide index values. This behavior indicates that the solvent type and the time of contact with the oil can promote lipid oxidation more readily and expressively, contributing to accelerating the product deterioration. Regarding the carotenoid content, the solid–liquid extraction of pequi pulp oil using acetone showed the highest carotenoid concentration. According to Rodriguez-Amaya [64], the extraction behavior can be explained by the high solvent selectivity and the lower temperatures of the extractive process. Low temperatures avoid the thermal degradation of carotenoids.

The literature mentions the high percentage of bioactive compounds in the cold-pressed extraction of vegetable oils [49,65]. However, the steps between pequi fruit collection and oil commercialization can be long, interfering with the concentration and quality of carotenoids. The storage conditions, type of equipment, and fruit maturation stage also influence carotenoid quality. In addition, the data regarding the physicochemical properties of the almonds and pequi pulp oils obtained by different methods are still poorly reported in the literature. Thus, it is still challenging to evaluate the pequi products’ quality when comparing the different fruit processing methodologies.

Table 8 presents the effect of the extraction method on the physicochemical properties of oils from pequi almonds and pulp.

### 6.2. Fatty Acids

Secondary metabolites such as oleic, palmitic, myristic, palmitoleic, stearic, and linoleic acids, among others, are found in the chemical constitution of pequi oil, which makes it an attractive option for the functional food market [18]. Fatty acids give pequi oil characteristics suitable for crystallization and melting, which are essential in the manufacturing of foods and products with a melting point close to the temperature of the human body, 37 °C [50]. Pessoa et al. [18] and Cornelio-Santiago et al. [67] reported oleic acid as a monounsaturated compound that favors the reduction in triglycerides, LDL cholesterol, total cholesterol, and glycemic index in the fight against cardiovascular diseases.

The most commonly used technique to identify and quantify fatty acids in pequi oil is gas chromatography (GC) [18,35,37,68]. In GC, the extraction, analysis, and determination of fatty acid esters present in the oil are carried out via the interaction of the stationary phase (pequi oil) with a mobile phase (gas).

The fatty acid profiles of the pulp and almond oils (Table 9) from pequi obtained by different extraction techniques indicate the richness of pequi oil in unsaturated fatty acids, mainly oleic and palmitic acids, and other acids in smaller proportions. No significant differences were observed concerning the types and percentages of fatty acids in the pequi and almond oils, demonstrating similar profiles regardless of the oil extraction method. The minimal discrepancy can be explained by factors such as the cultivation region and the fruit maturation stage, which may differ in the samples evaluated [15,33].

## 7. Challenges in the Purification of Bioactives from Pequi-Based Products

Pequi fruits are highly nutritious and of great economic importance to families working with them; however, their cultivation, collection, and processing in *Cerrado* areas are still little explored and recognized, making it difficult to add value to the productive chain of pequi. The high content of carotenoids, phenolic compounds, and unsaturated fatty acids make pequi oil a potential source of natural dyes and other bioactive substances, with its primary use as an input by the food, pharmaceutical, and cosmetic industries. However, as described by Alves et al. [62], it is a challenge in the pequi production chain to establish logistics for transporting and conserving the fruit and derivatives that result in a nutritionally and economically attractive product.

In this context, the composition of bioactive compounds in pequi oil depends on the conditions used in the collecting, transporting, storing, and handling stages. Pequi has a complex plant tissue structure that protects the bioactive substances in its pulp and almonds. However, some initial steps of the oil extraction process, such as peeling, cooking, and the pressing of the fruits, promote the disruption of the cellular structure and increase the surface area. Therefore, the exposure of bioactive compounds to oxygen, light, and greater temperatures also increases, putting them in contact with oxidative enzymes, which provide a series of degradation reactions [62,69].

It is necessary to develop methods that allow the efficient bioactive extraction from pequi oil while preserving its stability. The new methodologies should promote the cell wall rupture, release, and transport of the bioactive compounds to the external environment under conditions of minimized degradation. The solvent, temperature, and extraction time influence the purification efficiency of carotenoids and phenolic compounds in pequi [63].

The process of obtaining pequi oil by solid–liquid extraction becomes faster and more efficient when performed at high temperatures; however, the main obstacle regarding pigment extraction from pequi oil is the low stability of carotenoids to light and temperature as carotenoids are thermosensitive. The concentration of β-carotene can be reduced by 50% in a sample exposed to light for 30 min [15,62].

The exposure to adverse conditions accelerates degradation reactions due to the instability of bonds responsible for the characteristic color of carotenoids. An extraction above 40 °C can degrade pigments, reducing the carotenoid concentration in the oilseed extract. In this sense, to obtain an oil fraction enriched in carotenoids, the extraction should be performed below 40 °C using nonpolar extracting solvents since the oil and carotenes are hydrophobic [63,64].

Avoiding oxidation and maintaining the stability of the carotenoid extracts are other challenges in obtaining natural pigments [62]. Even after extraction, carotenoids can suffer degradation if exposed to external agents, such as temperature, humidity, oxygen, and ultraviolet radiation [70]. Techniques to prevent the carotenoid oxidative process, such as microencapsulation, should be used to avoid its degradation. Microencapsulation isolates the carotenoids from the extract in a layer of an encapsulating agent, usually a polymeric material, that acts as a protective film to avoid the effect of inadequate exposure. This protective membrane breaks down under a specific stimulus, and the substance can be released at an ideal place or time [71].

In general, the bioactive compounds from pequi oil benefit human health. The study of their retrieval must involve the storage, transport, and processing of the fruit and its derivatives to preserve the integrity of bioactive compounds so that they can perform their function excellently.

## 8. Future Perspectives on the Valorization of Pequi and Its Derivatives

The growing demand for vegan products and natural sources of bioactive molecules with anti-inflammatory, photoprotective, and antitumor properties boosts the technological use of pequi and its derivatives [72,73]. Pequi has a high nutritional and phytochemical composition with food and medicinal properties. Pequi is called the gold standard of the Brazilian *Cerrado*, given the significant contributions to its cultivation region [74].

Pulp and almonds are the most commonly used parts of pequi. Pequi oil from pulp and almonds has great applicability in the food industry, mainly due to its high concentrations of unsaturated fatty acids, vitamins, and carotenoids. Thus, the pulp and almond oils from pequi are the most produced pequi derivatives. In this sense, pequi oil use has tended to grow, mainly due to the valorization of natural pigments to the detriment of artificial dyes, which have been associated with health problems. Pequi oil (*i*) can enhance color and enrich dairy products [73,75]; (*ii*) has crystallization and melting properties suitable for manufacturing cosmetic products because of the high concentrations of oleic and palmitic fatty acids [23,66,69]; (*iii*) has humectant, emollient, and antioxidant action, capable of nourishing skin and hair and protecting them from damage caused by ultraviolet radiation; (*iv*) has an anti-frizz effect due to its capillary action; (*v*) and contributes to the softness and hydration of the hair, acting as a protective film, which helps replenish the oleic content and reduce water loss [67,73].

Ombredane et al. [76] developed nanoemulsions based on pequi oil that could fight breast cancer. Colombo et al. [42] and Miranda-Vilela et al. [36] reported the effectiveness of pequi oil supplementation in minimizing side effects from chemotherapy and reducing DNA damage and lipid peroxidation in lung cancer models.

Other products from pequi are also valuable, such as sweets, liqueurs, and ice cream, which constitute a source of income for many families harvesting and processing fruits [13,20,41]. Additionally, pequi residues, such as bark and thorns, can also be used technologically. According to Cangussu et al. [26] and Leão et al. [2], flours with high concentrations of fibers, proteins, minerals, and antioxidants, such as carotenoids, flavonoids, and phenolics, can be obtained from the peel and thorns of pequi. Furthermore, studies developed by Ghesti et al. [77] and Martins et al. [78] proved that the pyrolysis of pequi peel allows the production of biochar, ketones, furans, acids, alcohols, and phenols, all of which are products with applications in the chemical industry. Therefore, the literature allows us to infer that pequi can be used in the medicine, food, cosmetics, and energy industries. Pequi farming can also promote a financial return to families cultivating and processing fruits.

## 9. Conclusions

The *Cerrado* biome presents, among its numerous natural riches, a great diversity of endemic species of high economic and social importance, especially for families whose livelihood depends on their cultivation, collection, and processing, such as pequi. The potential of using this species goes beyond food and culinary purposes since pequi oil and other pequi-based products have been proven to have high levels of carotenoids, fatty acids, and vitamins. These substances are essential for the proper functioning of the human body and can be utilized to produce food, medicines, and cosmetics. Carotenoids can also be used as substitutes for synthetic dyes. Thus, considering all these benefits, combined with the trend toward the consumption of vegan and functional foods, it is expected that the technological application of pequi will evolve in the coming years. However, the high degradability of dyes and the low level of exploration and recognition of the species make it challenging to extract carotenoids and other bioactive substances from pequi oil, making it difficult to add value to its production chain. Since the stability of natural pigments is compromised by external agents, such as temperature, water content, oxygen, and ultraviolet radiation, it is necessary to develop careful logistics to establish conditions for the transport, storage, and processing of fruit and its derivatives to preserve or minimize the degradation process of bioactive compounds. In addition, the purification process of bioactive compounds from pequi may be developed to maintain its nutritional and economic viability.

## Figures and Tables

**Figure 1 foods-12-01907-f001:**
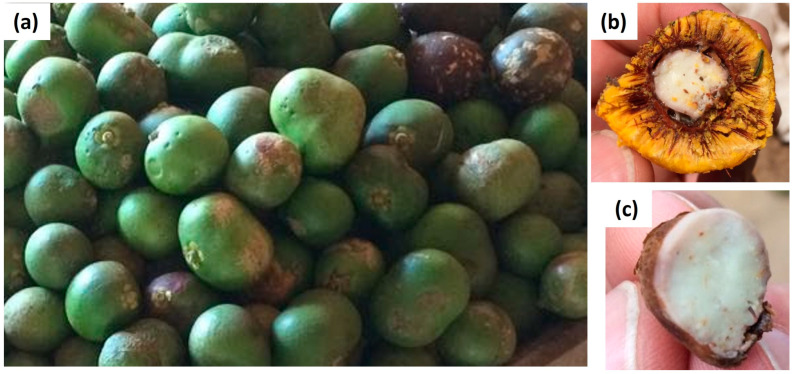
External (**a**) and internal parts of pequi (**b**) and (**c**) (source: the authors).

**Figure 2 foods-12-01907-f002:**
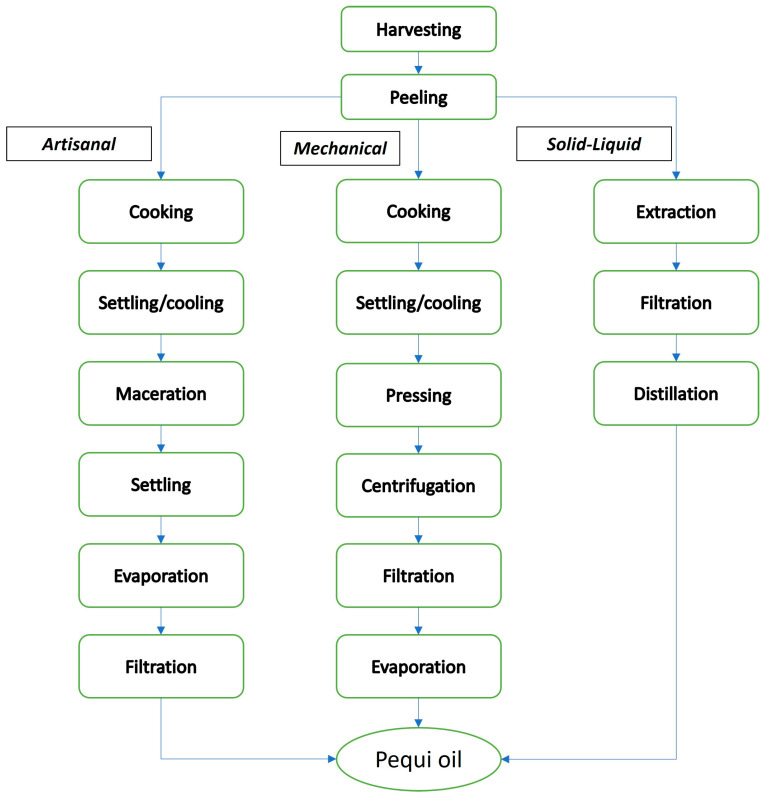
Process flowchart for obtaining pequi oil using artisanal, mechanical, and solid–liquid extractions (source: the authors).

**Figure 3 foods-12-01907-f003:**
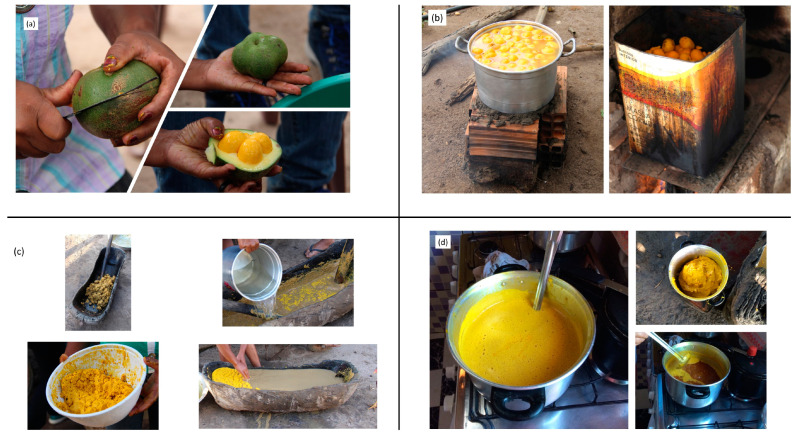
(**a**) Peeling of the fruits; (**b**) cooking of the fruits; (**c**) maceration of cooked fruits; (**d**) evaporation (source: the authors).

**Table 1 foods-12-01907-t001:** Chemical composition (g/100 g sample), vitamins (μg/100 g), energy value (kcal), carotenoids, and total phenolics (mg/100 g) of the almond and pulp from pequi cultivated in different Brazilian provinces (Provinces of Goiás, Mato Grosso, Minas Gerais, Tocantins, and Piauí).

	Pulp					Almond	Refs.
	Minas Gerais	Tocantins	Goiás	Mato Grosso	Piauí		
Proteins (g)	2.9 ± 0.49	1.74 ± 0.01	2.03 ± 0.25	2.63 ± 0.28	3.00 ± 0.13	1.74 ± 0.01	[17,19]
Ashes (g)	0.67 ± 0.05	0.57 ± 0.02	0.66 ± 0.02	0.62 ± 0.01	0.63 ± 0.01	0.57 ± 0.02	
Lipids (g)	27.13 ± 0.89	19.03 ± 1.61	18.95 ± 0.65	32.57 ± 0.78	33.4 ± 3.76	19.03 ± 1.61	
Carbohydrates (g)	7.02	6.78	6.4	3.6	11.45	6.78	
Moisture (%)	52.37 ± 1.34	61.56 ± 1.32	61.54 ± 1.22	53.2 ± 0.38	41.5 ± 2.00	61.56 ± 1.32	
Dietary fibers (total, g)	9.91 ± 0.17	10.32 ± 0.11	10.42 ± 0.18	7.43 ± 0.15	10.2 ± 0.2	10.32 ± 0.11	
Energetic value (kcal)	283.85	205.35	204.27	317.85	358.4	205.35	
Phenolics (total, mg/100 g)	215.87 ± 0.17	207.8 ± 0.09	221.2 ± 0.17	3.3 ± 0.14	209.00 ± 0.05	207.80 ± 0.09	
Carotenoids (total, mg/100 g)	18.7 ± 1.24	13.35 ± 0.47	11.1 ± 0.72	0.54 ± 0.01	7.25 ± 0.6	13.35 ± 0.47	
Mineral composition (mg/100 g)							
	Pulp			Almond			
Calcium	50–60			90.12 ± 0.71			[3,20]
Copper	240–400			--- ^a^			
Iron	830–1600			2.28 ± 0.13			
Phosphor	1.7–2.1			2214.46 ± 1.85			
Magnesium	130			452.11 ± 62.1			
Potassium	134			835.7 ± 15.46			
Sodium	210			--- ^a^			
Selenium (μg/100 g)	--- ^a^			1.4 ± 0.01			
	Vitamin composition (μg/100 g)						
	Pulp			Almond			
Vitamin A	20,000			650			[21]
Vitamin B1	30			10			
Vitamin B2	463			360			

^a^ Not reported by the authors.

**Table 2 foods-12-01907-t002:** Fatty acid (%) profiles of pequi pulp and almond cultivated in the Brazilian provinces of Goiás, Mato Grosso, Minas Gerais, Tocantins, and Piauí Adapted from Refs. [17,19] ^a^.

	Pulp					Almond
Fatty Acids	Goiás	Mato Grosso	Minas Gerais	Tocantins	Piauí	
Saturated						
Butyric (C4:0)	0.15		0.02			
Caprylic (C8:0)		0.02	0.03			
Capric (C10:0)	0.01		0.03			
Lauric (C12:0)	0.09	0.02	0.05		0.04	
Myristic (C14:0)	0.18	0.07	0.1	0.10	0.27	0.46 ± 0.01
Palmitic (C16:0)	28.05 ± 0.8	40.37 ± 0.11	36.78 ± 0.56	28.14 ± 0.28	35.17 ± 0.27	43.76 ± 0.04
Margaric (C17:0)	0.13	0.09	0.07	0.06		
Stearic (C18:0)	1.73 ± 0.02	2.3	2.07	1.39	2.25 ± 0.04	2.54 ± 0.06
Arachnid (C20:0)	0.2	0.14	0.23 ± 0.01	0.14 ± 0.01	0.23	0.02
Behenic (C22:0)	0.06	0.02	0.04			
Lignoceric (C24:0)	0.04	0.04	0.02	0.06		
Monounsaturated						
Palmitoleic (C16:1)	0.42 ± 0.01	0.96	1.18		1.03	1.23 ± 0.03
Heptadecenoic (C17:1)	0.15		0.08			
Oleic (C18:1) *cis*	60.4 ± 0.89	51.56 ± 0.15	50.72 ± 1.49	63.54 ± 0.35	55.87 ± 0.3	43.59 ± 0.16
Elaidic (C18:1) *trans*	0.03					
cis-vaccenic (C18:1) n7 ^b^					1.9 ± 0.08	1.38 ± 0.01
Eicosamonoenoic (C20:1)	0.26	0.14	0.46 ± 0.01	0.34		
Nervonic (C24:1)	0.14		0.08			
Polyunsaturated						
Linoleic (C18:2)	3.67 ± 0.04	1.01	1.13	1.61		
Docosadienoic (C22:2)	0.02		0.02			
Eicosatrienoic (C20:3) n3 ^b^	0.11		0.07			
Eicosapentaenoic (C20:3) n6 ^b^			0.04			
Arachidonic (C20:4)			0.03			
Eicosapentaenoic (C20:5)	0.38 ± 0.01		0.27			
Docosahexaenoic (C22:6)			----			
Total saturated	30.64	43.07	39.44	29.89	37.97	46.78
Total unsaturated	65.58	53.67	54.08	65.49	58.8	46.20

^a^ Missing values were not reported by the authors. ^b^ n corresponds to the position of the first double bond, counting from the terminal carbon (CH_3_). If n is omitted, its value is 9, the standard position for fatty acids.

**Table 3 foods-12-01907-t003:** Contents of carotenoids and phenolics from fresh and processed parts of pequi.

Carotenoids			
Fruit Part	Compound	Concentration (mg/100 g)	Refs.
Pulp	β-carotene	6.26–11.4	[33]
	Lycopene	1.12–2.08	
	Total carotenoids	6.75–11.34	
	β-Carotene	13.76 ± 0.01	[31]
	β-Carotene	0.07 ± 0.00	[32]
	All-trans-neoxanthin	0.23 ± 0.06	
	9-Cis-neoxanthin	0.08 ± 0.02	
	All-trans-violaxanthin	0.11 ± 0.02	
	9-Cis-violaxanthin	0.04 ± 0.01	
	Lutein	0.17 ± 0.04	
	All-trans-lutein	0.09 ± 0.02	
	9-Cis-antheraxanthin	0.06 ± 0.02	
	Antheroxanthin	0.34 ± 0.08	
	Zeaxanthin	0.29 ± 0.03	
	Mutatoxanthin	0.06 ± 0.05	
	Total carotenoids	1.73 ± 0.24	
	Antheroxanthin	9.24	[34]
	Zeaxanthin	7.91	
	Cryptoflavin	1.78	
	β-Carotene	1.47	
	ζ-Carotene	0.93	
	β-Cryptoxanthin	1.21	
	Mutatoxanthin	0.43	
	Total carotenoids	23.11	
Almond	Total carotenoids	0.03 ± 0.01	[35]
Pulp oil	β-carotene	54.98 ± 0.01	[31]
	Total carotenoids	49.99 ± 0.01	[36]
Almond oil	Total carotenoids	0.26 ± 0.00	[37]
	Total carotenoids	27.49 ± 0.34	[38]
Peel flour	β-Carotene	1.49 ± 0.18	[26]
	β-Cyptoxanthin	0.11 ± 0.01	
	Lutein	1.36 ± 0.16	
	Total carotenoids	3.49 ± 0.03	[2]
	Total carotenoids	3.38 ± 0.00	[39]
	α-Carotene	1.74 ± 0.27	[40]
	β-Carotene	1.85 ± 0.18	
	Lycopene	1.19 ± 0.12	
Pulp flour	α-Carotene	6.38 ± 0.11	
	β-Carotene	5.98 ± 0.18	
	Lycopene	4.07 ± 0.09	
Almond flour	α-Carotene	1.47 ± 0.28	
	β-Carotene	0.94 ± 0.13	
	Lycopene	1.36 ± 0.09	
Phenolic compounds			
Pulp	Gallic acid	78.30 ± 1.08	[41]
	Monogalloyl hexoside	2.11 ± 0.14	[32]
	Gallic acid	18.24 ± 1.7	
	Hexahydroxydiphenoyl-hexoside	2.04 ± 0.24	
	Coumaroyl-galloyl hexoside	0.45 ± 0.06	
	Coumaroyl quinic acid	0.27 ± 0.06	
	HHDP-dihexoside	1.74 ± 0.24	
	Digalloyl hexoside	1.84 ± 0.57	
	Galloyl-HHDP hexoside	2.06 ± 0.48	
	Ellagic acid hexoside	1.69 ± 0.2	
	Ellagic acid pentoside	0.93 ± 0.18	
	Ellagic acid deoxyhexoside	10.7 ± 0.94	
	Ellagic acid	10.4 ± 1.05	
	Methyl ellagic acid pentoside	0.82 ± 0.22	
	Methyl ellagic acid deoxyhexoside	1.05 ± 0.21	
	Methyl quercetin dihexoside	0.63 ± 0.11	
	Dimethyl ellagic acid pentoside	1.03 ± 0.16	
Almond	Gallic acid	211 ± 6	[35]
Pulp oil	Chlorogenic Acid	1.27 ± 0.12	[31]
Almond Oil	Gallic acid	0.39 ± 0.00	[37]
	Gallic acid	22,910,000 ± 165,000	[38]
Pulp Extract	Gallic acid	18.97 ± 1.93	[25]
	Ellagic acid	7.14 ± 0.07	
	*p*–Coumaric acid	0.31 ± 0.03	
Peel Flour	Gallic Acid	11.52–418.67	[26]
	Ellagic Acid	509.47–1630.66	
	Gallic Acid	20,893.73 ± 1462.14	
	Ethyl Gallate	2026.75–5205.90	
	Gallic Acid	9475.69 ± 12.74	[40]
Pulp Flour	Gallic Acid	402.21 ± 35.75	
Almond Flour	Gallic Acid	210.50 ± 34.95	

HHDP: Hexahydroxydiphenoyl.

**Table 4 foods-12-01907-t004:** Techniques applied to extract oil from pequi pulp.

Method	Operating Condition	Extraction Yield (%)	Refs.
Artisanal	Solvent: water; cooking: 100 °C; 1 h	19.37	[45]
Mechanical	Pre-drying: 60 °C; 24 h; pressing: 28 °C	22.4	[45]
Solid–Liquid	Solvent: ethyl ether; extraction time: 4 h	58.47	[45]
	Solvent: acetone; extraction time: 4 h	60.5	
Supercritical	Solvent: CO_2_; 40 MPa; 333.15 K	49	[46]

**Table 5 foods-12-01907-t005:** Solid–liquid oil extraction from pequi under different solvents and operating conditions.

	Solvent	Operating Conditions	Extracted Oil (%)	Refs.
Pulp	Hexane	69 °C	59.96	[51]
		50 °C; W = 10.37–1.36	42.5–59.4	[54]
		NR ^a^	49.6	[18]
		T = 50 °C, 22 Hz, 16 h, S-L = 1/10	60.2	[48]
	Ethanol	T = 78.37 °C	34.8	[51]
		NR ^a^	52.8	[18]
		T = 50 °C, 22 Hz, 16 h, S-L = 1/10	39.8	[48]
	Acetone	50 °C, 22 Hz, 16 h, S-L = 1/10	61.07	[48]
Almond	Hexane	55 °C	98.4	[35]
	Ethanol	55 °C	76.1	
	Acetone	55 °C	86.0	
	Isopropanol	55 °C	87.9	

S-L: Solid–liquid ratio (%); W: moisture (%, dry basis); ^a^ NR: Operational conditions were not reported.

**Table 6 foods-12-01907-t006:** The yield and operating conditions of the supercritical extraction of oil from pequi pulp.

Solvents	Operating Conditions	Extracted Oil (%)	Refs.
CO_2_	GT: 30 s, 333.15 K, 40 MPa, SR: 2.93 × 10^−4^ kg/s	49.0	[46]
CO_2_	GT: 50 s, 333.15 K, 40 MPa, SR: 2.93 × 10^−4^ kg/s	47.0	[46]
Propane	333.15 K, 15 MPa	43.7	[18]

GT: Milling time (s); SR: solvent rate (kg/s).

**Table 7 foods-12-01907-t007:** Advantages and disadvantages of the different methods of pequi oil extraction.

Extraction Method	Advantages	Disadvantages	Refs.
Artisanal	No sophisticated equipment. No generation of toxic waste. Oil with good quality. Oxidative enzyme inactivation via fruit cooking. Financial income for communities living off of fruit extraction.	Many hours of work. Low oil yield. High cooking temperatures can degrade nutraceutical compounds.	[45,51,55]
Mechanical	Low investment in energy and equipment. Preservation of the bioactive compound integrity. Simple operating methodology. No toxic waste generation.	Low yield. Considerable oil loss in the residual cake. Oil extract may contain fibers and/or impurities. Direct contact between fruit and metallic press parts can produce oil with free fatty acids.	[45,46,55]
Solid–Liquid	High efficiency. Minimal energy demand at low temperatures. Preservation of thermosensitive compounds.	Environmental and operational risks due to solvent toxicity. Possibility of bioactive thermal degradation. Higher operating costs. Requirement for oil separation from solvent.	[45,56]
Supercritical	High efficiency. High-purity oil extract free of residues. Shorter extraction time. Low temperatures. Bioactive compound preservation.	Expensive equipment.	[46,57]

**Table 8 foods-12-01907-t008:** Effect of the extraction method on the physicochemical properties of pequi almonds and pulp oils ^a^.

	Operating Conditions	Carotene	Acidity	PV	SV	Refs.
	Almonds					
Artisanal	Cooking: 100 °C, water, Almond/water: 1/3 *m*/*v*	0.1	0.55 ± 0.07	2.91 ± 1.44	---	[37]
Mechanical	Hydraulic pressing: 25 °C, 1 h, 9 ton-force	0.1	1.09 ± 0.68	0.85 ± 0.88	---	[37]
Solid–Liquid	Hexane, 90 °C, 6 h	---	4.94 ± 0.08	28.23 ± 0.05	206.10 ± 0.93	[66]
	Pulp					
Artisanal	Cooking: 100 °C, 1 h	25.0	1.44	0.760	214.36	[45]
Mechanical	Pre-drying: 60 °C, 24 h, Continuous pressing: 28 °C	26.9	5.44	1.07	225.1	[45]
Solid–Liquid (S-L:1:10, 16 h, 22 Hz)	Ethyl ether, 4 h	42.7	3.24	0.752	218.66	[45,48]
	Acetone, 4 h	38.9	2.71	0.725	217.3	
	Ethanol	29.9	---	197.23	195–198	
	Hexane	20.3	---	197.23	195–198	

^a^ Missing values were not reported. S-L: Solid–liquid ratio (%). Carotene: Total carotenoid in the oil (mg/100 g). Acidity: mg KOH/g. PV: Peroxide value (meq O_2_/kg). SV: Saponification value (mg KOH/g).

**Table 9 foods-12-01907-t009:** Fatty acid composition of pulp and almond oils from pequi obtained by solid–liquid (Soxhlet with hexane), supercritical (propane fluid), mechanical, and artisanal extractions ^a^.

		Pulp			
	Solid–Liquid [35]	Supercritical [18]		Mechanical [68]	
Fatty Acid	Content (%)				
Oleic (C18:1)	60.39	58.73		56.5	
Palmitic (C16:0)	33.87	34.95		38.11	
Stearic (C18:0)	1.71	1.74		2.61	
Palmitoleic (C16:1)	0.73	0.63		1.01	
Linoleic (C18:2)	2.22	2.84		0.96	
Linolenic (C18:3)	0.44	0.35		0.17	
Gadoleic (C20:1)	---	---		0.29	
Myristic (C14:0)	0.1	0.07		0.07	
Arachidonic (C20:0)	0.19	0.26		0.15	
	Almonds				
	Solid–Liquid	Artisanal		Mechanical	
	[66]	[1]	[37]	[1]	[37]
Oleic (C18:1)	50.2	54.97 ± 0.17	56.34 ± 0.15	54.39 ± 0.28	59.99 ± 0.03
Palmitic (C16:0)	42.3	34.92 ± 0.11	33.76 ± 0.03	34.78 ± 0.39	29.48 ± 0.00
Stearic (C18:0)	1.5	2.50 ± 0.04	2.62 ± 0.01	2.37 ± 0.07	2.44 ± 0.00
Palmitoleic (C16:1)	1.0	0.53 ± 0.01	0.59 ± 0.01	0.66 ± 0.05	0.66 ± 0.01
Linoleic (C18:2)	0.6	6.11 ± 0.11	5.74 ± 0.03	6.73 ± 0.13	6.48 ± 0.00
Linolenic (C18:3)	0.5	---	---	---	---
Myristic (C14:0)	0.2	0.35 ± 0.00	0.35 ± 0.00	0.37 ± 0.00	0.36 ± 0.00
Arachidonic (C20:0)	0.3	---	---		---
Erucic (C22:1)	---	---	0.60 ± 0.01		0.63 ± 0.01

^a^ Missing values were not reported.

## Data Availability

All data generated or analyzed during this study are included in the manuscript.

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
