# Peer review of "Potential Challenges of the Extraction of Carotenoids and Fatty Acids from Pequi (Caryocar brasiliense) Oil"

_foods, 2023, doi:10.3390/foods12091907_

Round 1

Reviewer 1 Report

the manuscript is quite interesting for readers. several comments are addressed to authors. I think the references of this manuscript are not enough (46 referecences)

1. paraphrase the sentence line 15 and 17

2. add the method of review, 

3.  is figure 1 original from authors? if not please add the reference

4. please move the the subchapter related carotenoid, pectin and other interest compounds of paqui in the beginning of manuscript

5. add the future perspective related the valorization of paqui in the one section

6. please make one table related the advantage and disadvantage of types of extraction.

Author Response

Reviewer 1: the manuscript is quite interesting for readers. several comments are addressed to authors. Thank you for reading our manuscript and for all the valuable comments and suggestions, which helped us improve the manuscript's quality.

I think the references of this manuscript are not enough (46 referecences). The number of references was increased to 69.

  1. paraphrase the sentence line 15 and 17.

The sentence "Natural sources of bioactive compounds target several scientific studies since these substances can promote countless benefits to the human body. The pequi is one of these natural sources that present a vast consumption of the fruit, in natura or processed, but it is still little explored economically" was replaced with "The pequi is one of the natural sources of bioactive compounds that present wide versatility of fresh or processed fruit consumption, but it is still little explored economically. Functional foods are the subject of diverse scientific research since, in addition to nourishing, they contain bioactive compounds capable of promoting various benefits to the human body."

  1. add the method of review. The method used was the narrative or traditional review. The information was added to the summary: "Therefore, the present narrative review aims to compile and critically…"
  1. is figure 1 original from authors? if not please add the reference. Yes, the original figure is from the authors.
  1. please move the the subchapter related carotenoid, pectin and other interest compounds of paqui in the beginning of manuscript Thank you very much for your comments. We added some information regarding carotenes in the beginning of the manuscript.
  1. add the future perspective related the valorization of paqui in the one section. The section "7. Future perspectives on the valorization of pequi and its derivatives" was added to the manuscript.
  2. please make one table related the advantage and disadvantage of types of extraction.Table 6 was added to the text.

Reviewer 2 Report

The study is interesting since pequi provides many bioactive compounds, but the detail is that the experiments are not properly represented.
The characterization of the samples does not show from which experimental results the results come.
There is no experimental design for the results shown, they do not mention if the tables shown are from their results or from what source they come from.
There is a lack of experiments, treatment of the samples, and a description of each analysis technique.
In the block diagrams shown, the time and temperature of the extraction methods are included.
Expand the conclusions of the work.

Include experiments and experimental design of all results

Author Response

Reviewer 2: The study is interesting since pequi provides many bioactive compounds, but the detail is that the experiments are not properly represented. Thank you for reading our manuscript and for all the valuable comments and suggestions, which helped us improve the manuscript's quality.

The characterization of the samples does not show from which experimental results the results come. The manuscript is a traditional (narrative) review of pequi oil. The introduction section was modified to "In this sense, the present work literature review (1) evaluates the methods for extracting oil from the pulp and almond of pequi and those for bioactive compounds from the oil, particularly pigments and fatty acids; (3) highlights the feasibility of exploiting this fruit species; and (4) presents the main challenges for turning more profitable the uses of pequi. It should be emphasized that the human diet highly influences the development of some autoimmune disorders and illnesses, such as cancer, nephropathies, and diabetes. Thus, including foods of high nutritional value containing bioactive compounds may be a way to prevent diseases (Salanță et al., 2020; Lisboa et al., 2020).

There is no experimental design for the results shown, they do not mention if the tables shown are from their results or from what source they come from. The manuscript is a traditional (narrative) review of pequi oil; thus, the tables were constructed with literature data.

There is a lack of experiments, treatment of the samples, and a description of each analysis technique. The manuscript is a traditional (narrative) review of pequi oil; thus, the information required was not presented.

In the block diagrams shown, the time and temperature of the extraction methods are included. Expand the conclusions of the work. The conclusion was changed to:

The Cerrado biome presents, among its numerous natural riches, a great diversity of endemic species of great economic and social importance, especially for families whose livelihood depends on their cultivation, collection, and processing, like the pequi. The potential use of this fruit species goes beyond food and culinary purposes since pequi oil and other pequi-based products have proven high levels of carotenoids, fatty acids, and vitamins. These substances are essential for the proper functioning of the human body and can be used to produce food, medicines, and cosmetics. Carotenoids can also be used as substitutes for synthetic dyes. Thus, considering all these benefits, combined with the trend toward the consumption of vegan and functional foods, it is expected that the technological use of this species will evolve in the coming years. However, the high degradability of dyes and the low level of exploration and recognition of the species make it challenging to extract carotenoids and other bioactive substances from pequi oil, making it difficult to add value to its production chain. Since the stability of natural pigments is compromised by external agents, such as temperature, water content, oxygen, and ultraviolet radiation, it is necessary to develop careful logistics to establish conditions for the transport, storage, and processing of fruit and its derivatives to preserve or minimize the degradation process of bioactive compounds. In addition, the purification process of bioactive compounds from pequi may be developed to maintain its nutritional and economic viability.

Include experiments and experimental design of all results. The manuscript is a traditional (narrative) review of pequi oil; thus, the information required was not presented.

Reviewer 3 Report

The work Potential and Challenges of the Extraction of Carotenoids and Fatty Acids from Pequi Oil (Caryocar brasiliense) constitutes an excellent review in this regard.

This work is clear and of interest to the scientific community.

Regarding the citations, most of them are less than 5 years old.

The conclusions are appropriate

It is very complete, just add that there are some sections that have not been translated into English (table 1 and table 2)

all the best

Author Response

Reviewer 3: The work Potential and Challenges of the Extraction of Carotenoids and Fatty Acids from Pequi Oil (Caryocar brasiliense) constitutes an excellent review in this regard. This work is clear and of interest to the scientific community. Regarding the citations, most of them are less than 5 years old. The conclusions are appropriate. It is very complete, just add that there are some sections that have not been translated into English (table 1 and table 2). 

Thank you for reading our manuscript and for all the valuable comments. Tables 1 and 2 present the names of some provinces in Brazil. For this reason, the names of the provinces Goiás, Mato Grosso, Minas Gerais, Tocantins, and Piauí were not translated into the English language. Tables 1 and 2 captions were changed to:

Table 1. Chemical composition (g/100g sample), vitamins (μg/100g), energy value (kcal), carotenoids, and total phenolics (mg/100g) of the almond and pulp from pequi cultivated in different Brazilian provinces (Goiás, Mato Grosso, Minas Gerais, Tocantins, and Piauí).

 Table 2. Fatty acid (%) profiles of pequi pulp and almond cultivated in the Brazilian provinces of Goiás, Mato Grosso, Minas Gerais, Tocantins, and Piauí [7,18].a

Reviewer 4 Report

Interesting work, based on new and up-to-date scientific reports. Collecting information in tables is helpful in illustrating the topic.

There are some considerations in the literature that can help you to improve the final version of this manuscript and were not described in the text.

It is well-known that common diseases and conditions such as obesity, diabetes, cancer, or autoimmune diseases can be accelerated or delayed depending on dietary options. Thus, healthy choices must include foods with nutrients and bioactive compounds, which, beyond the nutritional properties, can be used as an effective prevention strategy. In this regard, I kindly recommend the next paper to be consulted for the introduction section: DOI: http://dx.doi.org/10.5772/intechopen.91218; 10.1016/j.biochi.2020.09.002ff.

ffhal-02946034f

In my opinion, I think readers of this review paper may want to know about the future trends of pequi oil and other pequi-based products. It will be good if the authors could base it on the existing literature to give a more solid forecast about the trends.

Author Response

Reviewer  4: Interesting work, based on new and up-to-date scientific reports. Collecting information in tables is helpful in illustrating the topic. Thank you for reading our manuscript and for all the valuable comments and suggestions, which helped us improve the manuscript's quality.

There are some considerations in the literature that can help you to improve the final version of this manuscript and were not described in the text. It is well-known that common diseases and conditions such as obesity, diabetes, cancer, or autoimmune diseases can be accelerated or delayed depending on dietary options. Thus, healthy choices must include foods with nutrients and bioactive compounds, which, beyond the nutritional properties, can be used as an effective prevention strategy. In this regard, I kindly recommend the next paper to be consulted for the introduction section. The sentence added at the end of the Introduction section was:

A clear trend in the last decade has been related to adopting healthier lifestyles, focusing on healthy eating, and leading an active lifestyle (Wang et al., 2022). Altogether the fast development of the food industry, consumers possess new expectations regarding food and healthy diets (Salanță et al., 2020). For instance, due to the recent spread of the COVID-19 pandemic in 2020 and the consequential mandatory change of lifestyle (staying at home or limitations to exercise outside and at gyms), consumers have changed their eating habits and dietary supplementation habits (Hamulka et al., 2020; Laguna et al., 2020). Furthermore, the linkage between individuals with obesity and large significant increases in morbidity and mortality from COVID-19 has raised consumers' concerns regarding body weight (Laguna et al., 2020; Popkin et al., 2020). Therefore, the food sector companies must keep up with the consumers' interests and need when designing novel products.

Studies have shown that both dietary intake and circulating levels of β-carotene are inversely associated with the risk of all-cause mortality, mainly dietary β-carotene and that it has a protective effect on preventing non-communicable chronic diseases (Khalil et al., 2021). Moreover, numerous epidemiologic studies have been reported regarding the association between carotene and health outcomes, including Alzheimer's disease (AD), fracture, and various types of cancers (Yi et al., 2023). Carotenoids also play an imperative role in boosting body immunity, mainly through their effects on several antioxidant and anti-inflammatory pathways and components of the immune response. Thus, carotenoids may have roles as immune enhancers against COVID-19 and other emerging diseases and related syndromes (Khalil et al., 2021).

Besides the high carotenoid content, pequi oil also provides unsaturated fatty acids, which benefit human health and nutrition. Among them, polyunsaturated fatty acids (PUFA), like linolenic and linoleic acids, are essential because humans cannot produce them; as such, they must be obtained through food (Ferreira et al., 2020). Additionally, oils with a high content of monounsaturated fatty acids, such as oleic acid, can lower bad cholesterol levels and protect against heart disease (Almeida et al., 2021).

Considering the various technological processes available, processing pequi and its oil is a great alternative to produce healthier foods combined with the sustainable development of the Cerrado biome.

It should be emphasized that the human diet highly influences the development of some autoimmune disorders and illnesses, such as cancer, nephropathies, and diabetes. Thus, including foods of high nutritional value containing bioactive compounds may be a way to prevent diseases (Salanță et al., 2020; Lisboa et al., 2020).

DOI: http://dx.doi.org/10.5772/intechopen.91218 - Salanță, L. C., Uifălean, A., Iuga, C. A., Tofană, M., Cropotova, J., Pop, O. L., ... & González, C. V. (2020). Valuable food molecules with potential benefits for human health. The Health Benefits of Foods-Current Knowledge and Further Development, 1-45. http://dx.doi.org/10.5772/intechopen.91218

10.1016/j.biochi.2020.09.002ff.ffhal-02946034f - This reference was not found in literature.

Round 2

Reviewer 1 Report

the manuscript has already been improved.

Author Response

Thank you for the correction. A native English-speaking colleague performed a spell check of the manuscript's language, as highlighted in red.

Reviewer 2 Report

Potential and challenges of carotenoid and fatty acid extraction from pequi (Caryocar brasiliense) oil.
It seems to me that the authors made the suggested changes.
I have a doubt the authors did the extraction of components. Only some references are missing in the added paragraphs.

Author Response

English language and style are fine/minor spell check required. Thank you for the corrections. A native English-speaking colleague performed a spell check of the manuscript's language, as highlighted in red.

It seems to me that the authors made the suggested changes. I have a doubt the authors did the extraction of components. In the current narrative review, we mentioned the data available in the literature. Thus, our research group developed no experimental procedure to write the present narrative review. Thus, Our team is currently working on the extraction experiments of pigments from pequi, palm, and microalgae, but we did not report the results of our tests because we did not end the assays.

Only some references are missing in the added paragraphs. A revision was made, and the references cited were rechecked. The missed references were added to the text.
